

# Loss of inner kinetochore genes is associated with the transition to an unconventional point centromere in budding yeast

Nagarjun Vijay

Computational Evolutionary Genomics Lab, Department of Biological Sciences, Indian Institute of Science Education and Research, Bhopal, Madhya Pradesh, India

## ABSTRACT

**Background:** The genomic sequences of centromeres, as well as the set of proteins that recognize and interact with centromeres, are known to quickly diverge between lineages potentially contributing to post-zygotic reproductive isolation. However, the actual sequence of events and processes involved in the divergence of the kinetochore machinery is not known. The patterns of gene loss that occur during evolution concomitant with phenotypic changes have been used to understand the timing and order of molecular changes.

**Methods:** I screened the high-quality genomes of twenty budding yeast species for the presence of well-studied kinetochore genes. Based on the conserved gene order and complete genome assemblies, I identified gene loss events. Subsequently, I searched the intergenic regions to identify any un-annotated genes or gene remnants to obtain additional evidence of gene loss.

**Results:** My analysis identified the loss of four genes (NKP1, NKP2, CENPL/IML3 and CENPN/CHL4) of the inner kinetochore constitutive centromere-associated network (CCAN/also known as CTF19 complex in yeast) in both the Naumovozyma species for which genome assemblies are available. Surprisingly, this collective loss of four genes of the CCAN/CTF19 complex coincides with the emergence of unconventional centromeres in *N. castellii* and *N. dairenensis*. My study suggests a tentative link between the emergence of unconventional point centromeres and the turnover of kinetochore genes in budding yeast.

## INTRODUCTION

The increasing availability of genomic datasets across diverse species has allowed the use of comparative genomic approaches to study kinetochore evolution. Such comparative approaches have led to interesting evolutionary insights from species with unique kinetochores (*Drinnenberg & Akiyoshi, 2017*). The kinetochore is a complex of interacting proteins that have undergone frequent changes in gene content and exhibits strong signatures of adaptive evolution (*Malik & Henikoff, 2001*; *Talbert, Bryson & Henikoff,*

Corresponding author
Nagarjun Vijay,
nagarjun@iiserb.ac.in

*2004*; *Schueler et al., 2010*; *Drinnenberg, Henikoff & Malik, 2016*). Despite changes in gene content, the kinetochore in diverse species is known to consist of the inner kinetochore which is assembled close to the centromere DNA as a specialized form of histone (CENPA) and the outer kinetochore which interacts with microtubules. The inner and outer kinetochores are known to consist of several interacting protein complexes (see *Van Hooff et al., 2017* for more details). These kinetochore protein complexes show patterns of co-evolution with interacting components through correlated gene loss events (*Tromer, 2017*). The CCAN/CTF19 complex interfaces with both the inner and outer kinetochore acting as a foundation for kinetochore formation (*Hara & Fukagawa, 2017*; *Hinshaw & Harrison, 2019*). Surprisingly, large-scale systematic screening for kinetochore genes in the genomes of numerous eukaryotes has shown that the majority of the components of the CCAN/CTF19 complex are lost in many lineages (*Van Hooff et al., 2017*). Another unexpected observation was that the CENPA (CENH3 and CSE4 homolog) gene, which performs a central role in kinetochore function, was lost in many insect species. This potentially recurrent loss of the CENPA/CSE4 gene in insect species coincides with their transition to holocentricity (*Drinnenberg et al., 2014*). Interestingly, it has been shown that *C. elegans* holocentromeres are actually point centromeres that are dispersed at transcription factor hotspots (*Steiner & Henikoff, 2014*). Hence, it has been speculated that changes in the gene content of kinetochores might be functionally related to transitions in the centromere type (*Drinnenberg & Akiyoshi, 2017*).

Centromere sequences in the vast majority of species are repeat-rich regions that are thought to be defined epigenetically and/or through recognition of dyad rich regions with non-B-form DNA structures (*Kasinathan & Henikoff, 2018*). Despite advances in genome sequencing and assembly methods, the high repeat content of these centromeres makes it harder to assemble and study them. Hence, as a proxy to the study of centromere regions, comparison of tandem repeats across large phylogenetic distances have been performed to understand the evolution of centromere sequences (*Melters et al., 2013*).

One of the most interesting transitions during centromere evolution was the emergence of genetically defined point centromeres in budding yeasts (*Malik & Henikoff, 2009*). These ~150 bp long centromeres found in Saccharomycetaceae have been easier to study due to the lack of repetitive regions, availability of complete high-quality genome assemblies for multiple closely related species, gene knockout libraries and ease of experimental manipulation. Hence, budding yeast species are a popular system to study the evolution of centromeres and the set of proteins that interact with them (*Roy & Sanyal, 2011*).

In addition to changing their sequences, centromeres are also known to change their genomic positions without any change in the karyotype (*Montefalcone et al., 1999*). However, centromere evolution can also be accompanied by changes in the karyotype (*O'Neill, Eldridge & Metcalfe, 2004*). Within mammals, centromeres are known to have undergone multiple repositioning events (*Rocchi et al., 2012*). Centromere repositioning events seem to be fairly common and could have a role in driving speciation or, at the very least, have a non-negligible role in affecting the local genomic selection landscape.
Recently, it has been shown that in Naumovozyma, an unconventional centromere has come into existence at a location that is largely distinct from that expected based on synteny with other Saccharomycetaceae species (*Kobayashi et al., 2015*). The Naumovozyma genus has been suggested as a model for comparative genomics and study of adaptive evolution due to the various phenotypic differences compared to other yeast species (*Karademir Andersson & Cohn, 2017*).

In this study, I first screened the genome assemblies of eight Pre-WGD (Whole Genome Duplication) and twelve Post-WGD yeast species for the presence/absence of homologs of 67 kinetochore genes. I find evidence for the concurrent loss of multiple genes from the CCAN/CTF19 protein complex in Naumovozyma species and corresponding sequence divergence of the N-terminus region of the CENPA/CSE4 gene that interacts with the CCAN/CTF19 complex. In contrast to this, I see high levels of sequence conservation of the C-terminus region of the CENPA/CSE4 gene that mediates an interaction between the kinetochore and centromere. My analysis finds an association between gene loss events and the emergence of novel centromeres in Naumovozyma species.

## MATERIALS AND METHODS

### Gene presence/absence screening

I compiled a list of 67 kinetochore associated genes in the yeast *S. cerevisiae* by downloading genes annotated with the GO term kinetochore (GO:0000776) in Ensembl release 91. The orthologs of these genes in all twenty yeast species were identified from the Yeast Gene Order Browser (YGOB) (*Byrne & Wolfe, 2005*). I screened the genomes of all 20 yeast species for the presence of all 67 genes and identified eight genes that are (both copies lost in post-WGD) lost in at least one species (see Supplemental Material S1). Kinetochore genes are known to evolve at a very fast rate, making it hard to identify orthologs of these genes even in closely related species (*Van Hooff et al., 2017*). It is possible that these genes have evolved at a very fast rate making it unfeasible to establish homology of these genes. Fortunately, the YGOB provides not only the order of the genes but also the intergenic sequences between genes. Based on flanking genes with conserved synteny in other species, I identified the intergenic regions that correspond to the location of the missing genes (see Supplemental Material S2). I checked these intergenic sequences for the presence of open reading frames (ORFs) using the NCBI ORF finder program with default settings. The ORFs that were found in the intergenic regions did not show any homology (inferred using blastn and blastp search against NCBI's nucleotide collection and non-redundant protein sequences, respectively) to the genes that I have inferred to be lost. Based on the evidence from the identification of syntenic regions using YGOB and additional screening of intergenic regions I am confident of these gene loss events. Nonetheless, the presence of highly diverged copies of these genes in non-syntenic regions and genome assembly errors cannot be ruled out.

### Multiple sequence alignments of CENPA/CSE4 gene

The complete open reading frame of the CENPA (CENH3 and CSE4 homolog) gene was used for multiple sequence alignment. To ensure that the results I see are not the result of

alignment artifacts, I performed the multiple sequence alignment at the nucleotide sequence level and amino acid residue level using four different programs (Clustal omega (default settings in the webserver), M-coffee (command used and alignment scores are provided in Supplemental Material S3), MUSCLE (default settings in MEGA) and Guidance with PRANK as the aligner (command used is provided in Supplemental Material S3)). All the nucleotide and amino acid sequence alignments are provided as Supplemental Material S3. I investigated the multiple sequence alignments for evidence of lineage-specific patterns of selection in Naumovozyma species using the programs (RELAX, MEME, FEL and BUSTED) available in the HYPHY package (*Kosakovsky Pond, Frost & Muse, 2005*). The output files of these are also provided in Supplemental Material S3. The presence of additional sequences in the genus Naumovozyma was seen in all of the multiple sequence alignments that I have analyzed. It is possible that longer ORF's have been incorrectly annotated as the CDS for both Naumovozyma species. However, the second methionine codon in the CDS occurs at the 226th and 40th residue from the currently annotated start codon in *N. castellii* and *N. dairenensis*, respectively. If the second methionine is the correct start codon, the *N. castellii* protein would be just 39 residues. This suggests that the correct ORF is annotated in Naumovozyma species. To further rule out the possibility of erroneous annotation of shorter ORF's in non-naumovozyma species I extracted the genomic sequence found between genes flanking CENPA (CENH3 and CSE4 homolog) and searched them for ORF's using the NCBI ORF finder program with default settings (see Supplemental Material S4). I found that the longest ORF that could be identified in this sequence was the currently annotated ORF itself. This further supports the validity of the annotation and multiple sequence alignments generated by me. I used the amino acid alignment generated by MUSCLE aligner to calculate the sequence conservation score using the al2co program (*Pei & Grishin, 2001*) with default settings along with the-a flag to calculate nine measures of sequence conservation. Per base measures of sequence conservation are provided as Supplemental Material S5.

## Calculation of dyad density

Centromere sequences of the two Naumovozyma species were obtained from NCBI and YGOB browser for the remaining 18 species. The intergenic regions in Naumovozyma species orthologous to the old centromeres were extracted from YGOB. All these sequences and their locations are provided in Supplemental Material S6. The program *palindrome* from the EMBOSS package (*Rice, Longden & Bleasby, 2000*) was used to identify dyad symmetry (i.e., DNA sequences with base pairs that are inverted repeats of each other). The following settings were used for running the program "-minpallen 5 -maxpallen 100 -gaplimit 20-overlap". All the palindromes identified for each of the sequences is provided as Supplemental Material S7. The dyad density for each centromere sequence was calculated as the fraction of bases that are part of a dyad. GC content for each sequence was calculated as the fraction of GC bases. GC content and dyad density for each of the centromere sequences are provided in Supplemental Material S8.
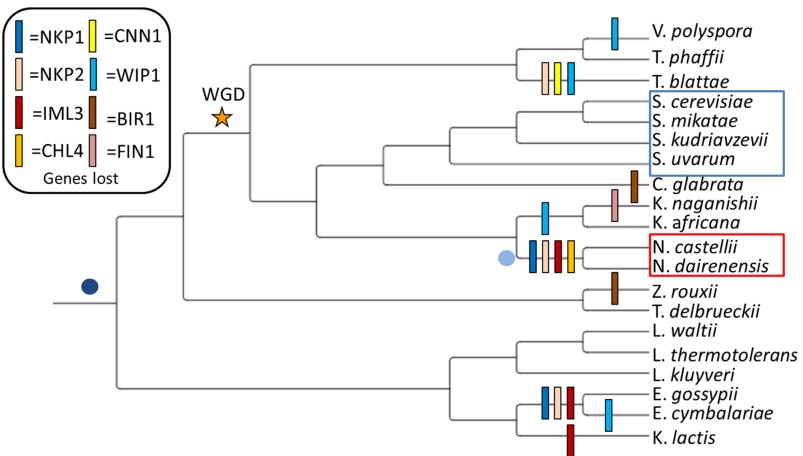

**Figure 1 Observed loss of kinetochore genes in budding yeast species.** Gene loss events have been mapped onto the widely accepted tree topology obtained from (*Gordon, Byrne & Wolfe, 2011*). The whole genome duplication event separating the Pre-WGD and Post-WGD species is denoted by a star. Point centromeres are common to all twenty budding yeast species and their emergence is depicted by a dark blue dot. Emergence of unconventional point centromere in Naumovozyma is depicted by a light blue dot. Sensu strictu yeast species are demarcated by a blue box and sensu lato species with neo-centromeres are demarcated by a red box.               

## RESULTS

### Patterns of kinetochore gene loss

My study system consists of eight Pre-WGD and twelve Post-WGD yeast species at varying evolutionary distances (see Fig. 1, phylogeny based on (*Gordon, Byrne & Wolfe, 2011*; *Feng et al., 2017*)). Based on screening of twenty yeast genomes for the presence of kinetochore genes I identified that eight genes (NKP1, NKP2, CENPL/IML3, CENPN/CHL4, CENPT/CNN1, CENPW/WIP1, FIN1 and BIR1) are lost at least once (see Fig. 1 and Supplemental Material S1). Intriguingly, five of the eight genes (i.e., NKP1, NKP2, CENPL/IML3, BIR1 and WIP1) that have lost both their copies in the Post-WGD species are also the ones that are lost in the Pre-WGD species (see Fig. 1). This hints at the dispensability of these genes over the course of evolution. In this study, I focus on the set of four genes (NKP1, NKP2, CENPL/IML3 and CENPN/CHL4) that are lost in Naumovozyma (a genus known to have novel point centromeres). As the name suggests, the Non-essential Kinetochore Protein genes NKP1 and NKP2 produce proteins that localize to kinetochores and when deleted produce viable single gene mutants (*Cheeseman et al., 2002*). However, both single gene mutants are known to show elevated rates of chromosome loss (*Fernius & Marston, 2009*). Knockout of both NKP1 and NKP2 only shows a moderate increase in chromosome loss (*Tirupataiah et al., 2014*). Despite being non-essential genes, at least one copy of both of these genes is found in 15 of the 20 yeast species screened in this study. A broader phylogenetic search for homologs of these genes has shown the repeated loss of these two genes in various taxa (*Tromer, 2017*). While it was known that NKP1 and NKP2 bind to COMA complex (*Hornung et al., 2014*), recent work has shown that NKP1 and NKP2 are positioned at the bottom of the CTF19 complex and form a four-chain helical coil along with OKP1 and AME1 at their

c-terminus (*Hinshaw & Harrison, 2019*). The main function of the NKP1 and NKP2 heterodimer is thought to be the stabilization of the COMA complex (*Schmitzberger et al., 2017*). The COMA complex is known to interact directly with the CENPA/CSE4 protein (*Fischböck-Halwachs et al., 2019*).

The CENPN/CHL4 (CHromosome Loss 4) single-gene knockouts are also viable but are known to show increased levels of chromosome loss, miss-segregation, and abnormal kinetochores (*Roy et al., 1997*). However, based on the chromatin state CENPN/CHL4 mutant cells are known to show two distinct (high and low) levels of mitotic mobility (*Roy & Sanyal, 2011*). This suggests that CENPN/CHL4 mutants can be compensated through changes in the chromatin state. CENPL/IML3 (Increased Minichromosome Loss) protein forms a stable heterodimer with CENPN/CHL4 protein (*Hinshaw & Harrison, 2013*). Although CENPL/IML3 and CENPN/CHL4 are known interactors, the phylogenetic distribution of CENPL/IML3 seems to be more restricted than that of CENPN/CHL4. The association of cohesin with the pericentromeric regions is ensured by the action of CENPL/IML3 and CENPN/CHL4. The lack of these two genes leads to reduced cohesin binding at the pericentromere that results in the miss-segregation of chromosomes (*Fernius & Marston, 2009*).

The function of the genes lost in Naumovozyma may be performed by genes that are found in Naumovozyma but are absent in *S. cerevisiae*. Naumovozyma species have 46 genes that are absent in the other 18 species analyzed in this study. To evaluate whether any of these genus specific genes could have taken over the role performed by the four missing genes, I identified homologs of each of these genes by performing blastp search against the non-redundant protein database with an *e*-value cut-off of $10^{-3}$. None of the identified homologs had a characterized role related to the kinetochore machinery. However, profile-versus-profile search based comparisons have identified that NKP1 and NKP2 are extremely similar to Mis12 and Nnf1 (*Tromer et al., 2019*). Since, one copy (NCAS0H00450 and NDAI0K02740) of the Nnf1 gene is present in each of the Naumovozyma species; it is possible that the functions of NKP1/2 are compensated.

## N-terminus divergence of CENPA/CSE4 gene in Naumovozyma

The detailed study of the interactions between kinetochore proteins in *S. cerevisiae* has led to a better understanding of their roles (*Measday et al., 2005*; *Baetz, Measday & Andrews, 2006*). However, the evolution of the kinetochore network across Eukaryotes has been shown to be a complex process that requires further investigation (*Van Hooff et al., 2017*). In the current study, I have focussed on well-studied genetic interactions of the selected genes that have lost both copies in Post-WGD yeast species. Chromatin Immuno-Precipitation (ChIP) of CENPA/CSE4 protein followed by sequencing of DNA fragments was used by *Kobayashi et al. (2015)* to identify all the CENPA/CSE4 binding sites in the *N. castellii* genome. Based on this experimental data for CENPA/CSE4 along with similar ChIP-seq data for the NDC10, NDC80, and CEP3 proteins the locations of the new centromeres in *N. castellii* have been validated (*Kobayashi et al., 2015*). The multiple sequence alignment of amino-acid sequences across the study species (see Supplemental Material S3) shows that the CENPA/CSE4 genes of *N. castellii* and
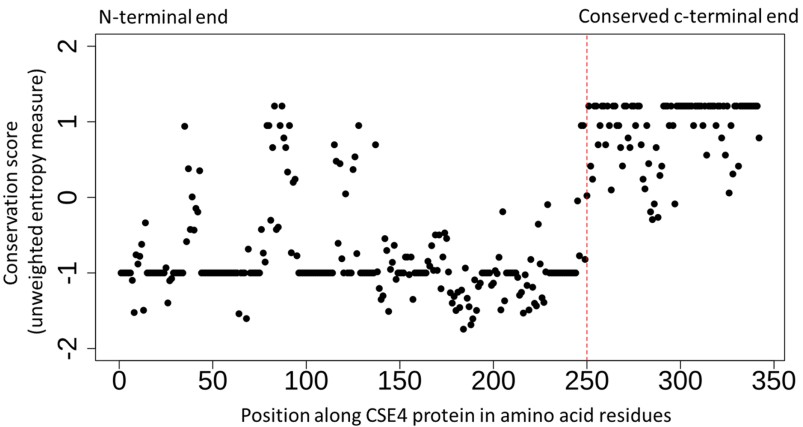

**Figure 2 Conserved c-terminus of CENPA/CSE4 contrasts with the diverse n-terminus.** Sequence conservation (unweighted entropy measure) along the length of the CENPA/CSE4 gene calculated using the al2co program is shown for each amino acid position. While the c-terminus is highly conserved, the n-terminus has very little sequence conservation. Refer to Supplemental Material S3 for actual multiple sequence alignments.                                         

*N. dairenensis* have a stretch of approximately ten amino acid residues at the very beginning of the CENPA/CSE4 gene that is absent in other species. It is known that the C-terminus of the CENPA/CSE4 protein binds to the centromeric DNA sequence, and the N-terminus interacts with the CCAN/CTF19 complex (*Chen et al., 2000*). The N-terminus region has overall reduced sequence conservation across all species compared to the C-terminus region (see Fig. 2). I also find evidence of significant relaxed selection in the *N. castellii* and *N. dairenensis* lineages using the hypothesis testing framework available in the RELAX program (version 2.1) on the multiple sequence alignments generated by MUSCLE. However, when the alignments generated using guidance with PRANK as aligner was used, the relaxed selection in the *N. castellii* lineage was not found to be statistically significant (see Supplemental Material S3). These differences in the inferences based on the multiple sequence aligner used are known to occur when the sequences are highly diverged (*Blackburne & Whelan, 2013*). I have shown earlier that genes can experience relaxed selection when the c-terminal end is increased in length due to change in the position of the stop codon (*Shinde et al., 2019*). Hence, the relaxed selection detected in the CENPA/CSE4 gene could simply be a result of drastic change in the length of the n-terminal sequence.

The N-terminus of the CENPA/CSE4 protein has been shown to play a role in the ubiquitin-mediated proteolysis of the CENPA/CSE4 protein (*Au et al., 2013*). More recently, sumoylation and ubiquitination of the N-terminus have been shown to be required to prevent the mislocalization of CENPA/CSE4 to non-centromeric chromatin (*Ohkuni et al., 2018*). The lysine 65 residue (K65) in the CENPA/CSE4 gene has been identified as the residue important for proper localization. Changes in the N-terminus region of CENPA/CSE4 gene could have led to changes in the post-translational modifications resulting in changes in the localization patterns and subsequent movement of centromeres in Naumovozyma.

Vijay
2020
10.7717/peerj.10085

```
>Scer_CEN1 Chr1 (151465..151582) [118 bp]
GTCACATGACATAATAATAAATAATTTTAAAAAATATAAAATATTTTTAATAGTTTTTAAATATTTTACAGTTTATTTTTTAAATTTATTTATATGTTTTTGTTTTCCGAAGCAGTCAA

GTCACATGACATAATAATAAATAATTTTAAAAAATATAAAATATTTTTAATAGTTTTTAAATATTTTACAGTTTATTTTTTAAATTTATTTATATGTTTTTGTTTTCCGAAGCAGTCAA

GTCACATGACATAATAATAAATAATTTTAAAAAATATAAAATATTTTTAATAGTTTTTAAATATTTTACAGTTTATTTTTTAAATTTATTTATATGTTTTTGTTTTCCGAAGCAGTCAA

GTCACATGACATAATAATAAATAATTTTAAAAAATATAAAATATTTTTAATAGTTTTTAAATATTTTACAGTTTATTTTTTAAATTTATTTATATGTTTTTGTTTTCCGAAGCAGTCAA

GTCACATGACATAATAATAAATAATTTTAAAAAATATAAAATATTTTTAATAGTTTTTAAATATTTTACAGTTTATTTTTTAAATTTATTTATATGTTTTTGTTTTCCGAAGCAGTCAA

GTCACATGACATAATAATAAATAATTTTAAAAAATATAAAATATTTTTAATAGTTTTTAAATATTTTACAGTTTATTTTTTAAATTTATTTATATGTTTTTGTTTTCCGAAGCAGTCAA

GTCACATGACATAATAATAAATAATTTTAAAAAATATAAAATATTTTTAATAGTTTTTAAATATTTTACAGTTTATTTTTTAAATTTATTTATATGTTTTTGTTTTCCGAAGCAGTCAA

GTCACATGACATAATAATAAATAATTTTAAAAAATATAAAATATTTTTAATAGTTTTTAAATATTTTACAGTTTATTTTTTAAATTTATTTATATGTTTTTGTTTTCCGAAGCAGTCAA

GTCACATGACATAATAATAAATAATTTTAAAAAATATAAAATATTTTTAATAGTTTTTAAATATTTTACAGTTTATTTTTTAAATTTATTTATATGTTTTTGTTTTCCGAAGCAGTCAA

GTCACATGACATAATAATAAATAATTTTAAAAAATATAAAATATTTTTAATAGTTTTTAAATATTTTACAGTTTATTTTTTAAATTTATTTATATGTTTTTGTTTTCCGAAGCAGTCAA

GTCACATGACATAATAATAAATAATTTTAAAAAATATAAAATATTTTTAATAGTTTTTAAATATTTTACAGTTTATTTTTTAAATTTATTTATATGTTTTTGTTTTCCGAAGCAGTCAA

GTCACATGACATAATAATAAATAATTTTAAAAAATATAAAATATTTTTAATAGTTTTTAAATATTTTACAGTTTATTTTTTAAATTTATTTATATGTTTTTGTTTTCCGAAGCAGTCAA

GTCACATGACATAATAATAAATAATTTTAAAAAATATAAAATATTTTTAATAGTTTTTAAATATTTTACAGTTTATTTTTTAAATTTATTTATATGTTTTTGTTTTCCGAAGCAGTCAA
GTCACATGACATAATAATAAATAATTTTAAAAAATATAAAATATTTTTAATAGTTTTTAAATATTTTACAGTTTATTTTTTAAATTTATTTATATGTTTTTGTTTTCCGAAGCAGTCAA
```

**Figure 3  Dyads identified in the *S. cerevisiae* centromere CEN1.** Inverted repeats were identified by the program palindrome from the emboss package. Each of the lines has the 118 bp sequence of CEN1 and the dyad region is highlighted in red color. The last line in the figure shows the regions (coloured brown) that are covered by at least one dyad. Dyad density is calculated as the fraction of bases that are covered by a dyad.               

## Dyad density at point centromeres

Dyad symmetry refers to the presence of inverted repeats or palindromes in the sequence. The inverted repeats identified on the centromere of *S. cerevisiae* are provided in Fig. 3 as an example. It has been shown that centromere regions are defined by the presence of Non-B-form DNA structures resulting from the presence of dyad symmetry in the nucleotide sequence (*Kasinathan & Henikoff, 2018*). While the new Naumovozyma centromeres are enriched for dyad symmetries and non-B-form DNA, the dyad symmetry was less and the SIST DNA melting and cruciform extrusion scores were lower in Naumovozyma (sensu lato, demarcated by a red box in Fig. 1) compared to sensu strictu species (demarcated by a blue box in Fig. 1 (*Pulvirenti et al., 2000*)). However, they utilized the contrast of sensu stricto vs sensu lato and ignore other species that have been phylogenetically placed closer to the sensu strictu species. Using a larger sample size (centromeres from various budding yeast species) I show that the GC content shows a significant negative correlation (Kendall's rank correlation coefficient tau of −0.59, *p*-value < 2.2E−16) with dyad density (see Fig. 4A). This correlation is reflective of the genome-wide pattern seen in *S. cerevisiae* (*Lisnić et al., 2005*).

The unconventional centromeres in the yeast species *N. castellii* and *N. dairenensis* are in most cases located at a different genomic locus compared to the point centromeres found in other budding yeast species such as *S. cerevisiae* (*Kobayashi et al., 2015*). This movement of centromeres has partly been attributed to chromosomal re-arrangements. I show that the GC content of the intergenic regions corresponding to the older (*S. cerevisiae* like) centromeres is very high and dyad density is low (see Fig. 4A; red colored circles). On the other hand the new unconventional centromeres that have been identified in Naumovozyma (see Fig. 4A; blue colored circles) are having higher dyad

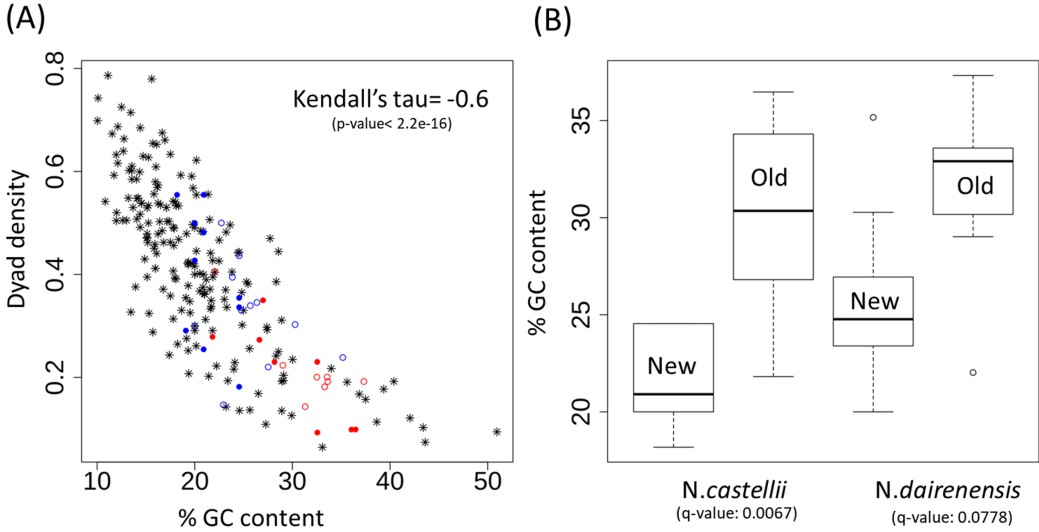

**Figure 4 Dyad density and GC content at the centromere regions.** (A) Negative correlation between GC content and dyad density at the point centromeres across budding yeast species. The filled circles represent *N. castellii* and unfilled circles show *N. dairenensis*. Red color is used for the intergenic regions corresponding to the older (*S. cerevisiae* like) centromeres. Blue color is used for the new unconventional centromeres that have been identified in Naumovozyma. Centromeres of the remaining budding species have the * shape and are colored black. (B) Boxplots comparing the GC content of old and new centromeres in *N. castellii* and *N. dairenensis*. The new (mean: 25.80, median: 24.77, min: 20.00 and max: 35.16) centromeres in *N. dairenensis* have a lower GC content than the older regions (mean: 31.58, median: 32.90, min: 22.03 and max: 37.32). Similarly, the new (mean: 21.36, median: 20.91, min: 18.18 and max: 24.55) centromeres in *N. castellii* have a lower GC content than the older regions (mean: 30.15, median: 30.35, min: 21.82 and max: 36.47). Pair-wise Wilcoxon test is used to compare the GC content of the old and new centromeres. The *q*-values are obtained based on holm multiple testing correction.

densities and lower GC content. I note that the pattern is the same for both Naumovozyma species (see Fig. 4A; filled circles for *N. castellii* and unfilled circles for *N. dairenensis*). The new (mean: 25.80, median: 24.77, min: 20.00 and max: 35.16) centromeres in *N. dairenensis* have a lower GC content than the older regions (mean: 31.58, median: 32.90, min: 22.03 and max: 37.32). Similarly, the new (mean: 21.36, median: 20.91, min: 18.18 and max: 24.55) centromeres in *N. castellii* have a lower GC content than the older regions (mean: 30.15, median: 30.35, min: 21.82 and max: 36.47). I used a pair-wise wilcoxon test with holm multiple testing correction (also see Fig. 4B) to compare the GC content of the old and new centromere regions and find a difference in both *N. castellii* (*q*-value: 0.0067) and *N. dairenensis* (*q*-value: 0.0778). This change in dyad density and GC content probably reflects a divergence that occurred after the change in the location of the centromere.

## DISCUSSION

Nucleotide sequence divergence at the centromeres themselves as well as in the coding sequence of proteins that recognize and bind to centromeres have been proposed as potential mechanisms for the build-up of post-zygotic reproductive isolation (*Borodin et al., 2001*). The paradoxical behavior of centromeres to evolve rapidly while still being inherited stably makes them good candidates for loci that contribute to the process of

speciation (*Henikoff, Ahmad & Malik, 2001*). However, the observation that Drosophila species produce fertile offspring despite the extensive divergence of the sequence of centromere binding proteins seems to contradict this idea (*Sainz et al., 2003*; but see *Thomae et al., 2013*). Computational identification and functional characterization of the effects of sequence divergence, loss, and duplication of the kinetochore genes in diverse species will help clarify the role of the centromere in facilitating reproductive isolation.

In this study, I provide evidence that suggests that both copies of four genes of the CCAN/CTF19 complex are lost in Naumovozyma budding yeast species that have transitioned to unconventional centromeres (*Kobayashi et al., 2015*). The loss of these CCAN/CTF19 complex genes is potentially mediated by changes in the N-terminus region of the CENPA/CSE4 gene (see Fig. 3B). Although I have no experimental data to suggest that the changes in the N-terminus region of the CENPA/CSE4 gene are a consequence of changes in the sequence of the centromere being bound by the C-terminus region, it is one potential scenario that could explain the observed association between gene loss and transition to unconventional centromeres. In contrast to the well-studied case of loss of CENH3 (CENPA/CSE4 homolog) gene in multiple insect lineages (*Drinnenberg et al., 2014*) and the multiple cases of exon gain and loss (*Fan et al., 2013*), I show that transition to a new centromere sequence might be sufficient for the loss of CCAN/CTF19 complex genes. Understanding the sequence of events involved in the loss of the CCAN/CTF19 complex genes and the emergence of the unconventional centromere in budding yeast species would help understand the loss of CCAN/CTF19 complex genes in various eukaryotic lineages (*Van Hooff et al., 2017*). The CENPA/CSE4 gene has additional sequence in Naumovozyma compared to other species suggesting that the unconventional centromeres might potentially be remnants of the ancestral state of point centromeres. I discuss a few other potential scenarios for the sequence of events that could have led to the association observed.

The dispensability of the genes that are lost in Naumovozyma is supported by the viability of *S. cerevisiae* knockouts and independent loss in the two Pre-WGD Eremothecium species. It is possible that initially, these genes were sequentially lost, and the emergence of novel centromeres was an adaptive response to compensate for the lost genes. Yet, the loss of three out of these four genes in Eremothecium species (*E. gossypii* and *E. cymbalariae*) does not seem to be associated with any noticeable changes in centromere type. The evolution of novel centromeres could just be one possible solution to the loss of these genes from the CCAN/CTF19 complex, and Eremothecium species might have come up with a different solution. Further phenotypic characterization of Eremothecium species might shed light on this.

It has been shown that loss of RNAi leads to a shortening of the centromeres and is an important determinant of centromere evolution in fungi (*Yadav et al., 2018*). The *N. castellii* species in addition to having unconventional point centromeres, have also been shown to have a functional RNA interference pathway (*Drinnenberg et al., 2009*). Despite having a role in fission yeast heterochromatin specification at the centromeres, siRNA's have not been found to have any centromere-specific role in *N. castellii* (*Kobayashi et al., 2015*). Nonetheless, it would be interesting to know what role the presence of RNAi

machinery in Naumovozyma might have had in the loss of CCAN/CTF19 complex genes. The *Drinnenberg et al. (2009)* study reports the gene expression fold change of ago1 knockout and dcr1 knockout in *N. castellii* compared to the wild type. However, none of the genes showing a fold-change greater than two are part of the kinetochore gene set.

The current study does not perform any experimental characterization of phenotypes across the twenty yeast species that would be caused by knocking out the genes that I identify as lost. However, kinetochore genes have been studied extensively in *S. cerevisiae*. Single and double gene knockouts have been phenotypically characterized in great detail (*Measday et al., 2005*). This functional data from *S. cerevisiae* can be extrapolated to other species by supplementing it with computational predictions. Such extrapolation would, of course, have to be experimentally verified by subsequent studies in the focal species. Hence, our study is merely focussed on demonstrating strong patterns of associations that suggest kinetochore network rewiring in Naumovozyma.

Stable separation of chromosomes into daughter cells requires proper functioning of the kinetochore machinery. Reduced efficiency in such segregation of chromosomes would result in reduced fertility. The high levels of gene turnover seen in kinetochore genes suggest frequent changes in the interactome of kinetochore. Such frequent changes in gene content could lead to differences in the interactome of closely related species. These differences in the interactome can have a prominent role in mediating speciation through reproductive isolation. Reconstruction of the sequence of events leading to the turnover of kinetochore machinery and centromere sequence at the molecular level in budding yeast species might serve as a good test case for understanding its role in speciation.

## CONCLUSIONS

Rapid changes in the genomic sequence of centromeres and associated kinetochore proteins between closely related species are thought to have an important role in speciation. Yet, it is not conclusively known whether the centromere sequence divergence occurs first or kinetochore proteins evolve to use a different centromere sequence. I looked at patterns of kinetochore gene loss in twenty species of yeast to identify major lineage-specific events. Interestingly, the loss of four genes of the CCAN/CTF19 complex coincides with the emergence of unconventional centromeres in *N. castellii* and *N. dairenensis*. I speculate that loss of these genes might have occurred subsequent to the divergence of the centromere sequences as their role might have been taken over by other genes. While my results cannot establish the sequence of events, the identified lineage-specific loss of kinetochores genes that are known to functionally interact serves as a molecular footprint of genetic changes that contribute to reproductive isolation between species.

### Funding

Nagarjun Vijay was awarded the Innovative Young Biotechnologist Award 2018 from the Department of Biotechnology and Early Career Research Award from the Department of

Science and Technology (both Government of India). The computational analyses were performed on the Har Gobind Khorana Computational Biology cluster established and maintained by combining funds from IISER Bhopal under Grant # INST/BIO/2017/019, Innovative Young Biotechnologist Award 2018 from Department of Biotechnology and Early Career Research Award from Department of Science and Technology. The funders had no role in study design, data collection and analysis, decision to publish, or preparation of the manuscript.

### Grant Disclosures
The following grant information was disclosed by the authors:
Department of Biotechnology, Government of India.
Department of Science and Technology, Government of India.
IISER Bhopal: #INST/BIO/2017/019.
Department of Science and Technology.

### Competing Interests
The author declares that he has no competing interests.

### Author Contributions
- Nagarjun Vijay conceived and designed the experiments, performed the experiments, analyzed the data, prepared figures and/or tables, authored or reviewed drafts of the paper, and approved the final draft.

### Data Availability
The raw data are available in the Supplemental Files.

### Supplemental Information
Supplemental information for this article can be found online at http://dx.doi.org/10.7717/peerj.10085#supplemental-information.

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
