# Peer review of "Loss of inner kinetochore genes is associated with the transition to an unconventional point centromere in budding yeast"

_PeerJ, doi:10.7717/peerj.10085_

## Round 0.1 · original submission · Minor Revisions

Your submission was evaluated by two independent reviewers. Both reviewers had positive, constructive comments whose consideration will improve the quality of your manuscript. I am looking forward at re-evaluating a new submission of your manuscript after you have incorporated reviewer's suggestions. One final important note: Please make sure that all and any data associated with this manuscript is readily available, either in the form of supplemental tables/figures and/or deposited to public databases. PeerJ aims to have the highest reproducibility standards.

Reviewer 1 ·

Basic reporting

The text is clear and unambiguous, but I recommend using the “CENP” nomenclature for proteins as well as the standard yeast names where possible. Text is unnecessarily long in places (see comments).

References are sufficient.

Article structure is OK. Figure quality is rather poor. Fig 2 is badly designed. Fig 3B is difficult to read. There is an error in Fig 1. See comments.

Self-contained. Hypothesis and results.

Experimental design

All items are OK. Need more details about how dyad density was calculated (Fig 4).

Validity of the findings

All OK.

Additional comments

This is an interesting manuscript. Previous work by Kobayashi et al (2015) found that, uniquely among the budding yeasts, two species in the genus Naumovozyma have point centromeres that deviate strongly from the standard point centromere consensus sequence established for Saccharomyces cerevisiae (with CDE I-II-III regions). Kobayashi et al also found that, in Naumovozyma, the locations of the centromeres on the chromsomes have changed. This led them to propose that in Naumovozyma, the new type of point centromeres displaced the old type (as opposed to a model where the old type gradually evolved into the new type). In the current manuscript, Vijay shows that 4 proteins in the inner kinetochore, which are usually quite well conserved, have been completely lost in Naumovozyma species. He speculates that these gene losses were either a cause or a consequence of the displacement of the point centromeres.

The methods used in the analysis are simple. Vijay used the YGOB comparative genomics browser to examine the locations and synteny of 68 kinetochore genes, and looked for species that have no homolog of each gene. He found that 4 inner kinetochore genes, IML3, CHL4, NKP1 and NKP2 are missing in two Naumovozyma species, and that 2 other genes CNN1 and WIP1 are missing in some other yeast species. IML3 and CHL4 interact with the centromeric histone-like protein CSE4 (a.k.a. CENP-A or CenH3) via the N-terminus of Cse4, and Vijay shows that the N-terminus in Naumovozyma is unusually long, indicating that it may have coevolved with the loss of the two proteins. He also shows that the new centromere sites in Naumovozyma have different properties than the previous centromere sites: they are more AT-rich and they have more dyad symmetry.

Overall, this manuscript contributes some useful insight into the change in centromere structure that occurred in Naumovozyma. In general, the conclusions are justified by the data.

Major comments

The Introduction section needs to provide a better introduction to the structure of the kinetochore for non-specialists. At the moment, it gives almost no background information at all. The nomenclature of kinetochore genes is complicated, because there are 70-80 proteins, in different (sub)complexes, and species-specific names are used in many organisms. The “CENP-” nomenclature system is an attempt to use standard names across multiple species (e.g. van Hooff et al 2017) and is widely used, but Vijay does not use it in this manuscript. I had to manually look up the facts that Iml3 is Cenp-L, Chl4 is Cenp-N, Cnn1 is Cenp-T and Wip1 is Cenp-W, so that I could then relate these proteins to the kinetochore structure figure in van Hooff et al (2017). Similarly, the names of complexes differ between species, for example I am not sure if CCAN means the same thing as Ctf19-associated complex.

NKP1 and NKP2 do not appear in van Hooff et al (2017) so the manuscript needs to explain where they fit into the structure of the kinetochore. The current text merely says that they “localize to kinetochores” (line 142). Is there evidence that they interact directly with CENP-A, L and N (which interact with each other)? If not, what part of the kinetochore do they interact with?

The Abstract should name the genes that are the focus of this paper: NKP1, NKP2, IML3, CHL4.

The text is too long in several places. Lines 194-213 contain a lot of discussion about genes that are in 2 or 1 copy in post-WGD species, but it seems to be inconclusive and only peripherally related to the main point of the manuscript which is gene losses.

Minor comments

Throughout the text, the author uses the word “intron” where he means “intergenic DNA”.

Results lines 135-137: It would be simpler to say that a group of 5 genes (name them) is lost in both pre-WGD and post-WGD species, with additional losses of 2 genes (name them) in pre-WGD species and 3 genes (name them) in post-WGD species.
Results line 139: Please explain why you only discuss 6 of the 8 genes that were lost in post-WGD species.

In Figure 1, I think that “Saccharomycetales” is a typo for “Saccharomyces”. But the topology of the tree is also incorrect. Naumovozyma and Kazachstania are sister genera, with Saccharomyces outside them.

Figure 2 and Supplementary Table 1 are difficult to read and understand. The author has simply downloaded the data file from the YGOB database, imported it into Excel, and colored some cells. For Figure 2, it must be possible to present the data in a clearer way (it is not necessary to show the gene names for each species). For Supplementary Table 1, most readers won’t know what they are looking at, and I don’t know why BIR1 and FIN1 were included.

Figure 3 highlights the unusual evolution of the N-terminus of Cse4. If it evolved under selection to compensate for the loss of the 4 inner kinetochore proteins, it could show evidence of positive selection on the sequence, which could be detected by using a program such as PAML. Did you investigate this?

In Figure 4, how was “dyad density” calculated, and what exactly does this term mean? I presume that It means inverted repeats, but what is the smallest size of dyad permitted? The Methods section does not say how it was calculated.

Reviewer 2 ·

Basic reporting

Overall the manuscript is of an acceptable standard. Points that could be improved are listed below:

- the word 'gene' is repeatedly used interchangeably with 'the gene's product', i.e. the encoded protein. For example lines 255-256: "... the genes that recognise and bind to centromeres...", and lines 41-42: "Kinetochore genes that form protein complexes...". This is scientifically incorrect, and should be amended.

- line 133 states that "eight Pre-WGD and twelve Post-WGD yeast species" were used for the analysis and that these are shown in Figure 1. However, the figure only lists a total of 14 species. In addition, the scale bar in Figure 1 labeled '20' is missing a unit, and the author needs to reference the source of the phylogenetic tree, or name the program and settings that were used to generate it.

- Figure 2 is too small to read. In addition, it would greatly benefit from clarifying the illustrated information. For example, the author could transpose the rows and columns, such that each gene of interest is given a column, and each species a row. The species could be grouped into pre- and post-WGD.

- Figure 4: the figure is very small, which could be mediated by making square rather than rectangular (i.e. stretch the y-axis). In addition the colours are to similar to distinguish between the groups (particularly comparing blue and black, or orange and brown). Also, the four CEN that only have (partially) inferred sequences should not be included in the analysis.

- The author might want to consider the order and / or headings for his results section. The paragraphs between lines 190 - 239 are under the heading N-terminal divergence of CSE4, and yet this protein is not mentioned once in these lines. Rather, at least some of these paragraphs would be better placed under the first results heading "patterns of kinetochore gene loss". Alternatively, additional heading(s) could be introduced to more accurately reflect the results that are described.

- lines 226-227: "... we are able to understand some of the variance seen in dyad density across budding yeast species". Based on the information provided, it is not clear how our understanding of the variance has improved. I suggest the author either clarifies this by clearly stating the insights that have been gained, or the author removes this statement.

- the manuscripts lists only a single author, and yet the manuscript consistently states "we". Unless additional authors are listed, I suggest replacing "we" with "I"

- I recommend using the past tense throughout the manuscript. In it's current state, the manuscript frequently uses the present tense, for example line 46 "Another unexpected observation is ...".

- line 41, the author quotes the kinetochore machinery to be a "Ship of Theseus". This is not a frequently used phrase and thus it's meaning unclear. I suggest either removing the phrase, or providing a meaning.

- the author refers to the 'origin' of point centromeres on lines 28, 29, 63, 91, 277, 285, 338 and 483. I would suggest to use the term 'emergence' rather than origin, which usually refers to a starting point.

- lines 177 (2x), 181, 183: the N- or C-terminal region of a protein is called N/C-terminus (i.e. the noun to terminal is terminus).

- line 220: the author refers to 'sensu lato' and 'sensu stricto' yeasts, but does not provide a definition. I suggest adding these labels to Figure 1, so as to clarify their meaning.

- line 143: "... proteins [...] produce viable single gene mutants...". This sentence should be clarified to state that "when deleted" the proteins produce mutants.

Experimental design

The methods section does not provide sufficient information that would allow readers to recreate the analysis. The author should specify how the checks for ORFs was performed and how the lack of homology was determined (lines 107 - 109). Importantly, the specific settings used for the multiple sequence alignments (listed on line 116) need to be provided.

Further minor suggestions are:

- line 98: include the actual GO term that was included, i.e. 'kinetochore', not just the associated number
- line 102: I suggest removing "Despite our precautions" as it is unclear what the author is referring to.

Validity of the findings

- lines 233 and 236: The author claims that there is a lower GC content in the new CEN in N. castellii. However, this is not apparent in figure 4. Instead the GC content appears comparable. The authors should determine the mean and median (including standard deviations) to support their statements. In addition, it would b greatly beneficial to include the genome-average mean and standard deviations for the GC content.

- The discussion and conclusion focus in particular on gene flow and speciation. However, the introduction is focussed on the CEN and kinetochore - gene flow is not mention, and the term 'speciation' occurs only ones in the introduction. The author could consider to adjust the emphases.

- lines 304 - 310: This discussion paragraph describes an analysis that was performed and as such should be relocated to the results section.

---

## Round 0.2 · accepted · Accept

Thanks for submitting your work to PeerJ. I apologize for the time it took to process this manuscript, but reaching reviewers was hard. I looked personally at your manuscript and I am attaching here a copy with few annotated observations. Please try to incorporate these changes before you submit your final version for publication.

One thing that I found 'strange' was the use of active first-person voice throughout your manuscript. While I would prefer the text to read in the passive voice, it is up to you to decide. A passive voice reads more 'neutral'. Regardless, I enjoyed the text.